# Molecular Dynamics Simulation on the Effect of Bonding Pressure on Thermal Bonding of Polymer Microfluidic Chip

**DOI:** 10.3390/polym11030557

**Published:** 2019-03-24

**Authors:** Mingyong Zhou, Xiang Xiong, Dietmar Drummer, Bingyan Jiang

**Affiliations:** 1State Key Laboratory of High Performance Complex Manufacturing, Central South University, Changsha 410083, Hunan, China; zmy_csu@163.com; 2Powder Metallurgy Research Institute, Central South University, Changsha 410083, Hunan, China; xiongx@csu.edu.cn; 3Institute of Polymer Technology, Friedrich-Alexander-University Erlangen-Nürnberg, Am Weichselgarten 9, 91058 Erlangen-Tennenlohe, Germany; drummer@lkt.uni-erlangen.de

**Keywords:** thermal bonding, PMMA, debonding, bonding pressure, molecular dynamics simulation, microfluidic chip

## Abstract

Thermal bonding technology is the most commonly used approach in bonding injection-molded microfluidic chips. Although the bonding mechanism is still under debate, the molecular dynamics (MD) method can provide insight into the bonding process on a macromolecular level. In this study, MD simulations for thermal bonding of PMMA substrate and cover sheet were performed. The molecule configuration and density distribution during the thermal bonding process were studied. The effects of bonding pressure on the equivalent strain, joining energy and diffusion coefficient were investigated. The debonding process was simulated to analyze the bonding strength and failure mechanism. Simulation results show that penetration mainly takes place near the interface area. Although the final density increases slightly with increasing pressure, the bonding interface is still insufficiently filled. The equivalent strain grows faster than that in the later stage because of the gap at the interface. The bonding pressure shows clear effects on the joining energy, diffusion coefficient and stress–strain behavior. Tensile failure occurs at the interface, with PMMA chains stretched between two layers. The majority of the change in potential energy is correlated with the change in non-bonded energy. At yield strain, the low-density defect at the interface weakens the tensile strength of bonded chip.

## 1. Introduction

The microfluidic chip, also referred to as the lab-on-a-chip system, is a novel tool for basic biological and chemical analysis within a short time period by controlling fluids in microchannel networks on a single chip. Depending on the desired application, which include electrophoresis, detection of pathogens, and DNA analysis, the design of the microchannel networks must be adapted [1,2,3], and the microfluidic chips must be precisely fabricated in order to meet the requirements of delivering a desirable result. The materials employed for the microfluidic chip are commonly silicon, glass and polymer, depending on the specific purpose and the most appropriate properties of these materials. Thermoplastic polymer is the most flexible material in the fabrication of the microfluidic chip, and can be processed via injection molding [4], hot embossing [5], laser machining [6], and other precision mechanical machining [7,8]. Among these methods, injection molding, as a replication technology, allows for the cost-efficient fabrication of polymeric devices with functional microstructure [9].

Following the injection molding process, the chip must be joined together to make sure it is firmly bonded and that there is no sign of leakage during the usage [10,11,12]. Thermal bonding of polymer chips is the most commonly used approach, because it allows the formation of microchannels with a uniform surface [13]. The proper control of bonding parameters such as bonding pressure and bonding temperature is critical to achieving a strong bond, while limiting the deformation of the microchannels on the substrate surface. In conventional thermal bonding experiments, bonding temperature is recommended to be near the glass transition temperature (*T*_g_) in order to achieve a good bonding. Bonding temperatures significantly higher than *T*_g_ cannot be applied in the thermal bonding process, as they would easily melt the microchannel pattern on substrate [13,14]. However, to date, the bonding mechanism of polymer is still under debate, even though molecule diffusion or adsorption theory can explain certain phenomena in polymer bonding with valid causes. An understanding of the polymer bonding mechanism based on the macromolecular level details is therefore of growing importance.

Molecular dynamics (MD) simulations can provide valuable insight into the interfacial behavior and mechanical property of polymer materials at a molecular scale, giving a view of dynamic evolution [15,16,17,18,19]. It is feasible to analyze the interdiffusion of polymer molecules across the interface and adsorption caused by molecular bonding, such as dipole-dipole interaction, van der Waals force and chemical bonding. Yokomizo et al. [20] used the coarse-grained MD method to study the effect of the molecular orientation of two polymer flow fronts on the interfacial structure. Yang et al. [21] investigated the surface welding process with thermally induced bond exchange reactions in an epoxy system and analyzed the penetration depth of polymer chains across the interface. Ge et al. [22] simulated the thermal welding of polymers to explore the increase in mechanical strength from interfacial entanglement with the welding time. Although MD simulation on thermal bonding of polymer is rarely reported, previous MD studies on thermal welding of polymers can provide useful references for thermal bonding analysis, because both processes are aimed to joining materials together based on thermal technology.

In this study, an atomistic model for thermal bonding of polymer substrate and cover sheet was constructed. Before bonding simulation, the glass transition process was investigated and the value of *T*_g_ was calculated for the following work. The configuration of combined system and corresponding distribution in mass density during thermal bonding process were studied. Next, the effects of bonding pressure on the equivalent strain, joining energy and diffusion coefficient were comprehensively investigated. Finally, a debonding process was simulated to analyze the bonding strength and to understand the failure mechanism. The output will give a useful reference in parameter determination and provide an understanding in bonding mechanism and failure behavior of the thermal-bonded polymer chip.

## 2. Simulation Models and Methods

### 2.1. Materials and model Constructing

Poly(methyl methacrylate) (PMMA) is a commonly used transparent thermoplastic in polymer microfluidic chips. In this simulation, isotactic PMMA was selected as the thermal bonding material for both the cover sheet and the substrate. Both PMMA layers were constructed in a square box with dimensions of 7.5 nm (length) × 7.5 nm (width) × 5.9 nm (height). There was a total of 40 molecule chains in the box, and the degree of polymerization was set as 50. The initial density of the PMMA layer was set to be 1.18 g/cm^3^ at a temperature of 298 K. Afterward, energy minimization and a subsequent annealing treatment were utilized to optimize the conformation of molecule structure. The thermal bonding model, also named the combined system, was then built by assembling one PMMA layer as the substrate and the other PMMA layer as the cover sheet. A schematic diagram of the thermal bonding process and the corresponding atomistic model for MD simulation are shown in Figure 1. Additionally, the periodic boundary conditions in the model were selected in order to approximate a large system.

### 2.2. Force Field and Simulation Procedure

In this simulation, a consistent valence force field (CVFF) was adopted to describe the intermolecular and non-bonded interactions between atoms in the polymer layer. The force field consists not only of bond stretching, angular bending and torsion potential, but also of non-bonded interactions, including Lennard-Jones (12-6) and Coulomb potentials. The non-bonded interaction between the cover sheet and substrate layer was also described by Lennard-Jones potential, with the cut-off distance for the non-bonded interaction being 1.25 nm.

Because of the time-temperature superposition principle, the calculated result of *T*_g_ by MD simulation is supposed to be higher than the experimental value [23]. Hence, the glass transition process needs to be recalculated by utilizing the Fox-Flory free-volume method. In this simulation, a single PMMA layer with an isotactic configuration was gradually cooled down from 505 to 205 K in a constant particle number, pressure and temperature (NPT) ensemble. The pressure in this system is set as a constant of 1 MPa, with a time step of 0.2 fs and a total simulation step of 100,000 steps.

Prior to the thermal bonding simulation, both PMMA layers were pre-heated at a constant temperature for 10 ps to reach a homogeneously heated state. In this present work, the bonding temperature was set to be the same as the calculated *T*_g_ of PMMA. After that, a bonding pressure ranging from 0.5 to 6 MPa in the z direction was applied to both layers by adding an external force to each atom. The bonding process was undertaken over a total of 150 ps in a constant particle number, volume and temperature (NVT) ensemble, with a time step of 1 fs. During the bonding simulation, periodic boundary conditions in both the x and y directions were used, while the non-periodic and shrink-wrapped boundary condition in the z direction was set so that both layers could be shrunk. Additionally, the interaction between atoms from the top of the cover sheet layer and the atom at the bottom of the substrate layer could be efficiently avoided by the non-periodic boundary condition. 

The bonded model was cooled down to room temperature (293 K) in 10 ps and subsequently relaxed at 293 K for another 10 ps. The simulation result was regarded as the initial model for the debonding investigation. After simulating the bonding process, a uniaxial tension simulation was performed to test the adhesive strength of PMMA interfacial bonding in NPT ensemble. Atoms in the bonded chip were displaced with a uniform velocity in the z direction so that the effective engineering strain rate was 10^11^ s^−1^. Due to the general limitations of MD simulation, the strain rate was orders of magnitude higher than the experimental value. The pressure in the x and y directions was set as zero in the uniaxial tension simulation. The debonding process was simulated with a time step of 0.1 fs and a maximum strain of up to 100%. All the simulations mentioned above were performed in a computer cluster by LAMMPS 64-bit version (25th June 2014) [24], an open-source molecular dynamics package distributed by Sandia National Laboratories, Livermore, CA, USA.

## 3. Results and Discussion

### 3.1. Glass Transition Temperature

*T*_g_ is one of the most critical thermodynamics properties of amorphous polymers. Drastic changes in physical properties, including viscosity and hardness, occur during the glass transition process. Figure 2 shows a graph of the volume of isotactic PMMA as a function of temperature. The intersection of the two lines fitted to both ends of the data would give an estimate of *T*_g_. The calculated value of *T*_g_ is 398.8 K, which is similar to those found in previous studies with other force fields [25]. Due to the huge difference in the cooling rates between molecular dynamics simulation and experimental research, the calculated *T*_g_ is higher than the experimental value. As mentioned above, the thermal bonding temperature of thermoplastic microfluidic chip should be near the *T*_g_ point. Therefore, the bonding temperature in the following study was set at a constant value of 398 K.

### 3.2. Bonding Process

Figure 3 shows snapshots taken during the thermal bonding process at simulation times of 0, 50, 100 and 150 ps, in which the bonding pressure in the z direction was 4 MPa. Both the cover sheet and the substrate layer were independently color-coded according to the atom height. As shown in Figure 3, there is a clear approach of atoms towards each other in both PMMA layers. Due to the presence of an obvious gap between the two layers, efficient bonding hardly takes place at the beginning. Here, the bonding interface was defined at a height of 11.4 nm in the z direction. As displayed in Figure 4a, the interface between the two layers is marked by a sharp drop in mass density of PMMA. When the bonding pressure is continued, atoms in both layers are gradually pushed towards the bonding interface. An interfacial interaction between these two layers will soon occur, because the non-bonded interaction is related to every atom located within a cut-off distance of 1.25 nm. At 100 ps, the interspace in the middle is mostly filled with PMMA molecules, generating a bonding interface with growing depth. It is found that atoms from the cover sheet and substrate layers slightly penetrate into each other. The combined system is further compressed at 150 ps, with a final penetration depth of 2.4 nm, which demonstrates that the penetration mainly takes place near the bonding interface area. Only a few atoms in the glassy PMMA system can cross over the interface from one layer to the other. Although the two PMMA layers are inserted into each other in large areas at the interface, the average density at the interface is about 0.8–1.0 g/cm^3^, which is relatively lower than that in the nearby bulk of each layer, as indicated in Figure 4b. By increasing the bonding pressure to 6 MPa, the final mass density increases slightly. However, the phenomena of insufficient filling at the interface can be still found after the bonding process. It is demonstrated that the glassy PMMA suggests poor flowability with restricted bonding pressure and time.

### 3.3. Bonding Pressure Dependence

Thermal bonding processes under various pressure conditions of 0.5, 1, 2, 4 and 6 MPa were simulated to investigate the bonding mechanism and the effect of bonding pressure. It is known from previous studies that bonding pressure that is too high will result in a large deformation in the microchannel on the substrate surface, which is not desirable during the thermal bonding process. In our previous studies, it was demonstrated that the microchannel decreased linearly in height and width as the bonding pressure increased. The maximum deformation in microchannel height reached 14.375%, with a bonding pressure of 3 MPa and a bonding temperature close to the *T*_g_ point [13,26].

Figure 5 shows the final snapshots of the thermal bonding of the PMMA substrate and cover sheet at 150 ps. With a bonding pressure of 0.5 MPa, the gap in the middle of the combined system still obviously exists at the final time, which means that the non-bonded interaction at the interface is hardly formed. With increasing bonding pressure, the gap is gradually filled, and more contact areas between these two layers are observed. To investigate the deformation during the thermal bonding process, the equivalent strain was computed by calculating the height difference of the system in the z direction. Figure 6 displays the variation in the equivalent strain as a function of bonding time with different bonding pressures. In general, a trend of higher strain rate with higher bonding pressure can be observed. Except for the case with 0.5 MPa, the equivalent strain in the combined system increases and then gradually appears to be stabilizing with increasing bonding time. Because of the gap in the middle of the combined system (shown in Figure 3), the deformation grows faster than that in the later stage. The final strain s at 150 ps are approximately 0.27 (6 and 4 MPa), 0.23 (2 and 1 MPa) and 0.15 (0.5 MPa). Regarding the situation with 0.5 MPa, we increased the bonding time to 500 ps (curve not shown). It was found that the equivalent strain was almost invariable after 400 ps, finally reaching 0.22 in the end. Further research could probably include the systematic study of the effect of bonding time during the thermal bonding process.

To elaborate the bonding mechanism of the PMMA substrate and cover sheet during the thermal bonding process, the diffusion and joining energy at the interface were introduced to reveal the inner reason. Here, the diffusion coefficient was calculated by following the Einstein relation in the mean square displacement (MSD) curve, as shown in Equation 1 and Equation 2. To obtain the diffusion coefficient, those data in the MSD curve that possessed a linear relation with simulation time were used to calculate the corresponding slope.
(1)$$MSD=\langle {\left|\vec{r_{i}}\left(t\right)-\vec{r_{i}}\left(0\right)\right|}^{2}\rangle$$
(2)D=limt→∞16t〈|ri→(t)−ri→(0)|2〉
where $\vec{r_{i}}$ is the reference position of each atom and *t* is the simulation time. The MSD–time profile under various bonding pressure conditions is shown in Figure 7. It can be seen by comparing Figure 7a,b that the majority of atom displacements come from movements in the z direction. In addition, the variation in total MSD–time profile is almost consistent with the profile in MSD in the z direction. With a higher bonding pressure, the MSD tends to rise more rapidly and to achieve a higher level during the thermal bonding process. When the pressure was 6 MPa, for instance, initial exponential-like growth and the following fluctuation in the MSD profile at the final stage were observed. It was demonstrated that the bonding pressure mainly contributed to the atom movement towards the middle of the system thanks to the free volume theory. As shown in Figure 8, the diffusion coefficient under a bonding pressure of 0.5 MPa is $26.7\times 10^{-6}\;{\mathrm{cm}}^{2}/s$, whereas the diffusion coefficient under 6 MPa is much higher, reaching a maximum value of $157.6\times 10^{-6}\;{\mathrm{cm}}^{2}/s$.

The chemical bonding theory proposes that the bonding mechanism could be attributable to intermolecular forces such as the non-boned interaction energy between the substrate and the cover sheet. In this study, the joining energy at the interface was calculated by the following Equation:(3)$$E_{joining}=-E_{inter}=-\left(E_{combined}-\left(E_{substrate}+E_{cover}\right)\right)$$
where $E_{combined}$ is the energy of the combined system, while $E_{substrate}$ and $E_{cover}$ are the energies of the substrate and cover sheet layer, respectively. As indicated in Figure 8, the joining energy at the interface increases with applied pressure. The joining energy reaches a maximum value of 912.4±121.3 kcal/mol with a bonding pressure of 6 MPa. However, when the bonding pressure is 0.5 MPa, the joining energy is only 12.8±9.3 kcal/mol. This means that the non-boned interaction at the interface is rarely generated, and the combined system is almost not bonded at 150 ps. In particular, in terms of 0.5 MPa, the final energy is increased to 508.5±183.1 kcal/mol with a longer bonding time of 500 ps, still far below the joining energy with a bonding pressure of 4 MPa, in which case the energy is 702.9±97.7 kcal/mol.

### 3.4. Debonding Process

The debonding process on the bonded chip was undertaken to further investigate the bonding strength. Since the combined system with a bonding pressure of 0.5 MPa did not bond effectively, we therefore used the system with a bonding pressure of 0 MPa for the uniaxial tension process instead, as a reference to compare the influence of bonding pressure. In addition, a deformation simulation on the substrate layer with homogeneous PMMA was also performed to obtain the stress–strain response of the pure PMMA material in tensile test.

Figure 9 shows snapshots of the deformation of bonded chip at the final strain of 100%. Regardless of the applied pressure, failure occurs at the interface during the debonding process. Stretched PMMA chains can be clearly observed in the middle of bonded chip. When the bonding pressure is relatively low, such as at 1 and 2 MPa, the crack width is larger due to the relatively lower interfacial interaction energy. When the cover sheet and substrate layer were bonded with 6 MPa, at least three PMMA chains were highly stretched, because of the adhesion between two layers, thus bridging the cover sheet and the substrate layer. It is demonstrated that the non-bonded energy contributes to the thermal bonding at the interface, enhancing the bonding strength after bonding.

Figure 10 illustrates the potential energy change of debonding process for the bonded chip. As mentioned above, the potential energy consists of bond stretching energy (*E*_bond_), angular bending energy (*E*_angle_), torsion potential energy (*E*_dihedral_) and non-bonded interaction energy (*E*_non-bonded_). In general, the majority of the change in potential energy in the bonded PMMA layer is correlated with the change in non-bonded energy. The non-bonded energy decreases sharply during the initial stage of the debonding process, and then increases gradually to its maximum. Both bond stretching energy and angular bending energy decrease steadily and then increase slightly thanks to the elastic recovery. However, there is little change in torsion potential energy with increasing strain, approximately remaining constant. With a higher bonding pressure, the changes in non-bonded energy, bond stretching energy and angular bending energy are relatively larger. As shown in Figure 9, the cover sheet and substrate layer are almost separated from each other after debonding. This means that the joining energy at the interface tends to be zero, the change of which is also included in the total change in non-bonded interaction energy.

As shown in Figure 11, the influence of bonding pressure on the stress–strain behavior was also investigated. Initially, the stress increases almost linearly with the strain, which is regarded as the elastic deformation in the tensile test. It can be observed that the elastic modulus of the bonded chip is relatively lower than that of the PMMA substrate. This is mainly because the PMMA substrate has a homogeneous density, without any bonding interface along the thickness direction. It is demonstrated in Table 1 that the bonding pressure has a clear effect on the stress–strain behavior, which can be revealed by comparing the peak yield stress and the corresponding strain at zero stress. With a bonding pressure of 6 MPa, the peak yield stress reaches 210.4 MPa, whereas the yield stress of the bonded chip with 1 MPa is 184.0 MPa. When the bonding pressure is greater, more strain is required to realize a zero stress, which means the bonded PMMA can bear a higher strain. Although the system with 0 MPa is not actually bonded, the yield stress can be still observed because the strain in this MD simulation is applied to every atom in both layers. By comparing the results obtained at 0 and 6 MPa, the peak yield stress increased by 106.4 MPa, nearly 35% of the yield stress in the PMMA substrate. The authors tried to increase the bonding pressure to 8 MPa, and the corresponding peak yield stress turned out to be 206.7 MPa. The results show little difference when comparing the yield stress with bonding pressures of 4 and 6 MPa. This means that when the bonding pressure is above 4 MPa, the peak yield stress in that case is mainly influenced by other processing parameters like bonding temperature and bonding time, rather than the bonding pressure. Additionally, the yield stress occurs at a strain of approximately between 0.06–0.09, relatively lower than that of the substrate, as shown in Figure 11 and Table 1. At the final stage, the stress–strain relations in all the systems are close to zero. The only exception is the behavior in the PMMA substrate, in which case the substrate is still tied together by molecules. It is known that PMMA materials have a low elongation at failure under actual conditions. The substrate in this MD simulation would also break into two parts soon after the final strain of 100%. Similar behavior in PMMA material was also reported in MD tension simulation using the DREIDING force field [27].

To further explore the deformation and configuration during debonding, the average mass density at yield point was studied. As seen in Figure 12, the mass density shows no significant sign of distribution difference for all curves, except at the interface region, where an obvious drop in density profile is observed. The phenomenon becomes more distinct as the bonding pressure gets lower. When the bonding pressure is 1 MPa, the lowest density is only 0.39 g/cm^3^, less than half that of the bonded chip with 6 MPa. By comparing the density of the bonded chip with 4 MPa (shown in Figure 4b), it can be known that the average density at the interface becomes lower in general. It is demonstrated that the low-density defect at the interface weakens the tensile strength during the debonding simulation. As shown in Figure 12, the decrease in mass density also implies that the yield first occurs at the interface.

## 4. Conclusions

In this paper, MD simulations for thermal bonding of PMMA substrate and cover sheet were performed to study the effects of bonding pressure on the equivalent strain, joining energy and diffusion coefficient. Finally, the debonding process was then simulated to analyze the bonding strength and the failure mechanism. The main conclusions of this present work are:

(1) Due to the huge difference of cooling rates between MD simulation and experimental research, the calculated *T*_g_ was higher than the actual value.

(2) Penetration mainly takes place near the bonding interface. Only a few atoms in glassy PMMA systems can cross over the interface from one layer to the other. Although the final density increases slightly with increasing pressure, the bonding interface is still insufficiently filled due to the poor flowability of glassy PMMA.

(3) The equivalent strain grows faster than that in the later stage because of the gap in the middle of the combined system. Initial exponential-like growth and a subsequent fluctuation in the MSD curve during the final stage were observed in most cases. With a higher bonding pressure, the MSD tends to rise more rapidly, showing a higher diffusion coefficient. The joining energy at the interface increases with applied pressure, reaching a maximum of 912.4±121.3 kcal/mol with 6 MPa. 

(4) Tensile failure occurs at the interface, with PMMA chains being stretched between two layers. The majority of the change in potential energy is correlated with the change in non-bonded energy. Bonding pressure shows a clear effect on the stress–strain behavior. With a bonding pressure of 6 MPa, the peak yield stress increased by 106.4 MPa, nearly 35% of that in PMMA substrate. The low-density defect at yield strain that is observed at the interface weakens the tensile strength of bonded chip.

## Figures and Tables

**Figure 1 polymers-11-00557-f001:**
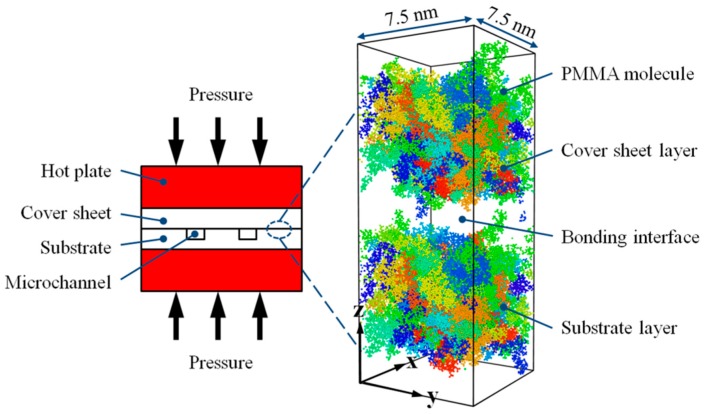
Schematic diagram of the thermal bonding process and the corresponding atomistic model for MD simulation.

**Figure 2 polymers-11-00557-f002:**
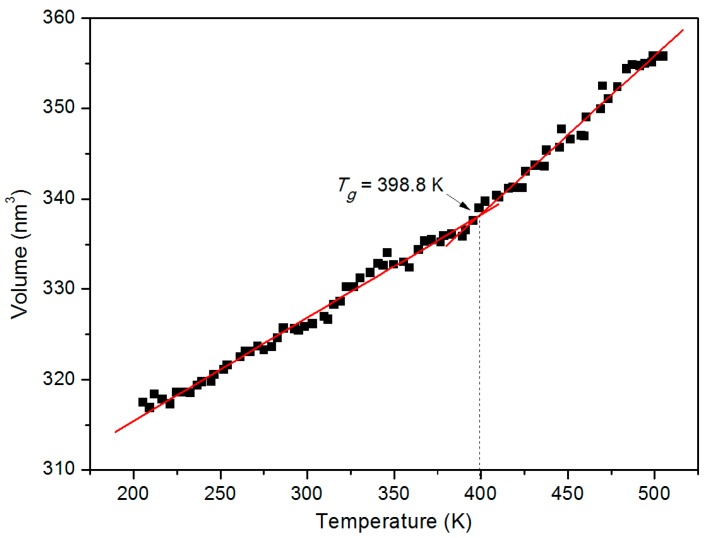
Volume–temperature curve in a single PMMA layer, with temperature gradually cooled down from 505 to 205 K.

**Figure 3 polymers-11-00557-f003:**
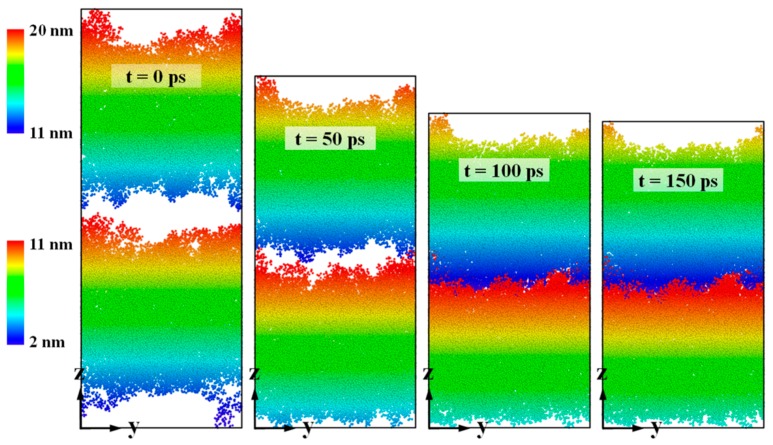
Snapshots of thermal bonding of the PMMA substrate and the cover sheet at simulation times of 0, 50, 100 and 150 ps, with a bonding pressure of 4 MPa in the z direction.

**Figure 4 polymers-11-00557-f004:**
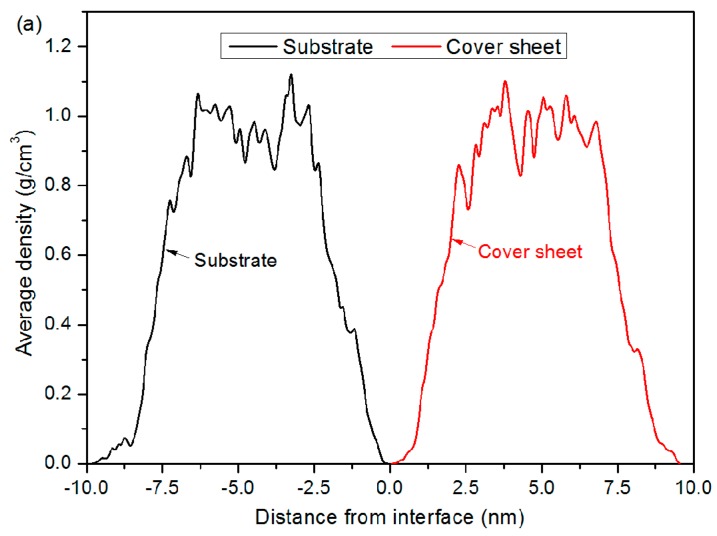
Average density profiles of the cover sheet, substrate layer and combined system along the z direction (**a**) before and (**b**) after the bonding process, with a bonding pressure of 4 MPa. z = 0 was defined as the center of combined system, which was at a height of 11.4 nm.

**Figure 5 polymers-11-00557-f005:**
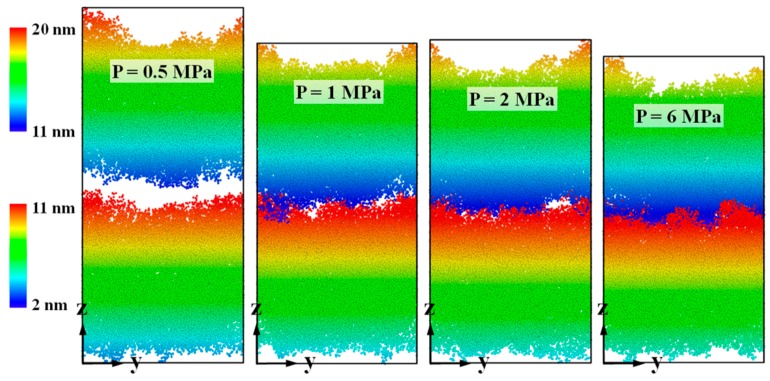
Snapshots of the thermal bonding of PMMA substrate and cover sheet at a simulation time of 150 ps, with bonding pressures of 0.5, 1, 2 and 6 MPa in the z direction.

**Figure 6 polymers-11-00557-f006:**
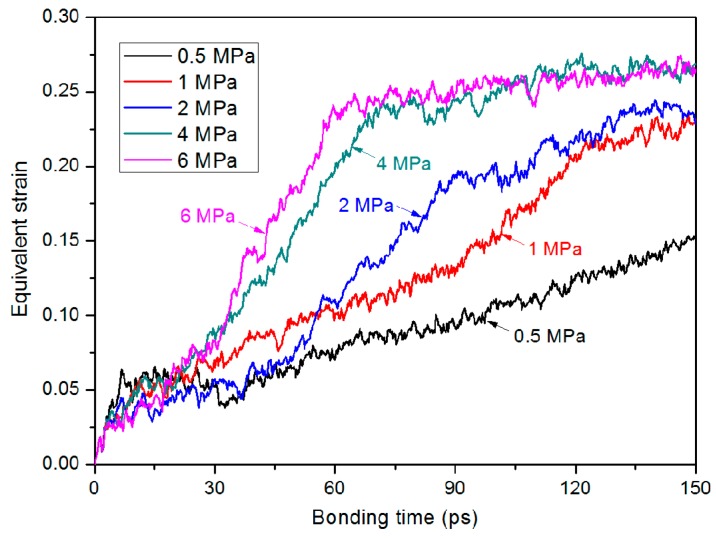
Comparison of the equivalent strain –time profiles under various bonding pressure conditions.

**Figure 7 polymers-11-00557-f007:**
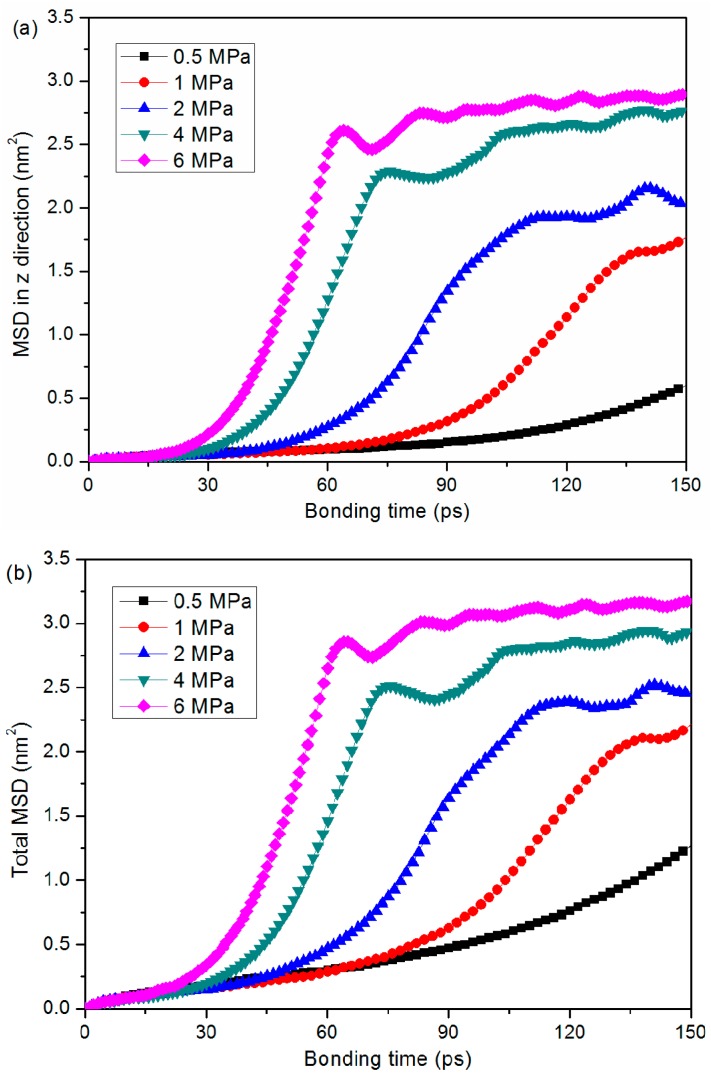
Comparison of the MSD–time profiles under various bonding pressure conditions. (**a**) MSD in the z direction and (**b**) the total MSD.

**Figure 8 polymers-11-00557-f008:**
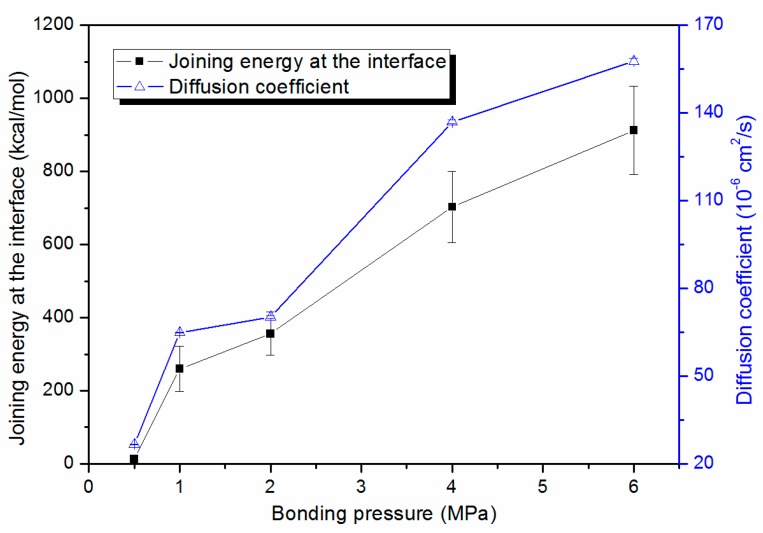
The effects of bonding pressure on joining energy at the interface after bonding and the diffusion coefficient during the thermal bonding process.

**Figure 9 polymers-11-00557-f009:**
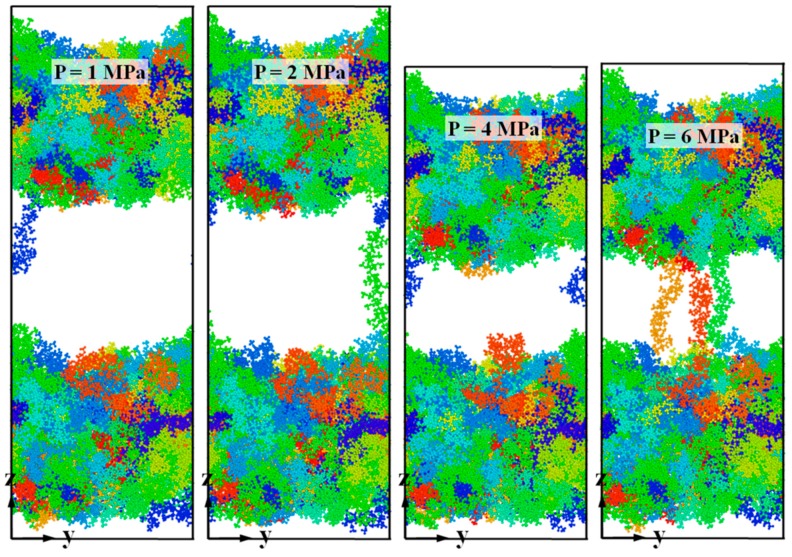
Snapshots of the deformation of the combined PMMA system at the final strain of 100%. The systems were bonded under various pressure conditions of 1, 2, 4 and 6 MPa in the thermal bonding process.

**Figure 10 polymers-11-00557-f010:**
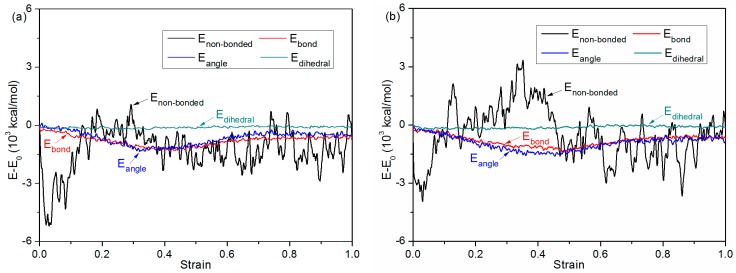
Potential energy decompositions of debonding process for the combined system under various pressure conditions of (**a**) 1 MPa, (**b**) 2 MPa, (**c**) 4MPa and (**d**) 6 MPa.

**Figure 11 polymers-11-00557-f011:**
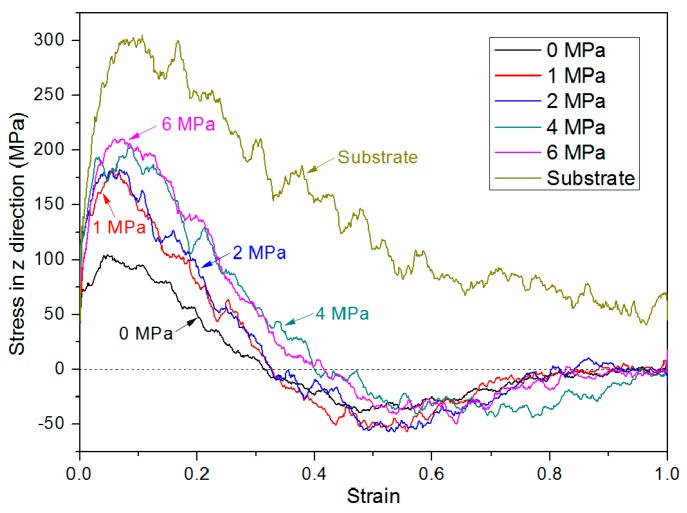
Stress–strain responses for different combined systems that were thermal bonded under various pressure conditions of 1, 2, 4 and 6 MPa in the z direction.

**Figure 12 polymers-11-00557-f012:**
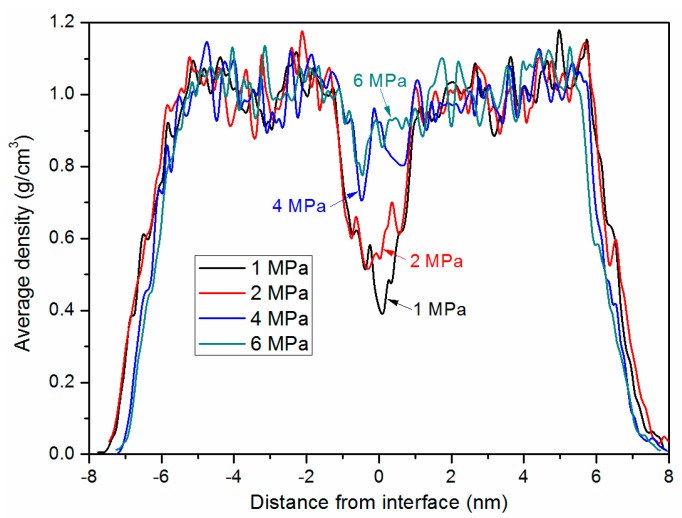
Average density profiles of the combined system along z direction at yield strain.

**Table 1 polymers-11-00557-t001:** Peak yield stress, yield strain and strain at zero stress of substrate and combined system under various bonding pressure conditions.

Simulation model/Bonding Pressure (MPa)	Substrate	0	1	2	4	6
Peak yield stress (MPa)	305.3	104.0	184.0	182.9	203.3	210.4
Yield strain	0.106	0.048	0.057	0.068	0.085	0.070
Strain at zero stress	N/A	0.314	0.327	0.326	0.400	0.416

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
