# Peer review of "Molecular Dynamics Simulation on the Effect of Bonding Pressure on Thermal Bonding of Polymer Microfluidic Chip"

_polymers, 2019, doi:10.3390/polym11030557_

Round 1

Reviewer 1 Report

This is a very well written paper in a very interesting field. MD simulation  in the bonding/adhesion field may be the way to go to understand this complex field. The English is good. I only found one typo:  3.3 bonding pressure dependance (Bonding should in caps). After this minor change, the paper can be published.

Author Response

Response to Reviewer 1 Comments

Point 1: This is a very well written paper in a very interesting field. MD simulation in the bonding/adhesion field may be the way to go to understand this complex field. The English is good. I only found one typo: 3.3 bonding pressure dependence (Bonding should in caps). After this minor change, the paper can be published.

Response 1: Thanks for the reviewer's comment. The typo error in line 177 has been modified in the revised manuscript (marked in red).

Reviewer 2 Report

The manuscript investigates an interesting application of MD simulation in trying to understand the complex problem of PMMA bonding/adhesion behavior. 

The technical part is very well organized, it can be considered for publication in MDPI Polymers after addressing the following questions:

Have the authors compared the simulation results (e.g. stress-strain behavior) with experimental data and observations?       

Have the authors tried beyond 6MPa? Would it further improve peak yield stress?  Or will it cause too much deformation? Some more results and discussion will be helpful in clarifying this. 

Author Response

Response to the Comments

The manuscript investigates an interesting application of MD simulation in trying to understand the complex problem of PMMA bonding/adhesion behavior. The technical part is very well organized, it can be considered for publication in MDPI Polymers after addressing the following questions:

Point 1: Have the authors compared the simulation results (e.g. stress-strain behavior) with experimental data and observations?

Response 1: Thanks very much for the good suggestion. Because of the huge difference in processing time and scale between MD simulation and experimental study, the strain rate is orders of magnitude higher than the experimental values due to the general limitation of MD simulations. As a result, there are differences between MD simulations and experimental results. Therefore, the comparison work is not discussed in this manuscript. Nevertheless, both MD simulations and experiments show that the bonding pressure and bonding time have big impacts on bonding strength. And some parameter ranges in this simulation are selected according to the previous experimental studies (in Ref. 13 and Ref. 26). Such description can be found in the revised manuscript (in Line 124-125 and Line 175-182).

Point 2: Have the authors tried beyond 6MPa? Would it further improve peak yield stress? Or will it cause too much deformation? Some more results and discussion will be helpful in clarifying this.

Response 2: Thanks for the advice. The authors have tried to increase the bonding pressure to 8 MPa and the corresponding peak yield stress turns out to be 206.7 MPa. The results show little difference by comparing the yield stress with bonding pressures of 4 MPa and 6 MPa. It means when the bonding pressure is above 4 MPa, the peak yield stress in that case is mainly influenced by other processing parameters like bonding temperature and bonding time, rather than the bonding pressure. Related explanation has been added to the revised manuscript (in Line 291-296, Section 3.4).